# Neurological Adverse Events Related to Immune Checkpoint Inhibitors: A Practical Review

**DOI:** 10.3390/ph17040501

**Published:** 2024-04-14

**Authors:** François Zammit, Emmanuel Seront

**Affiliations:** Institut Roi Albert II, Department of Medical Oncology, Cliniques Universitaires Saint Luc, Avenue Hippocrate 10, 1200 Brussels, Belgium; francois.zammit@student.uclouvain.be

**Keywords:** immune checkpoint inhibitors (ICIs), neurological immune-related adverse events (n-irAEs), myasthenia gravis, encephalitis, neurotoxicity, immunotherapy

## Abstract

The increasing use of immune checkpoint inhibitors (ICI) in cancer therapy has brought attention to their associated neurotoxicities, termed neurological immune-related adverse events (n-irAEs). Despite their relatively rare incidence, n-irAEs pose a significant risk, potentially leading to severe, long-lasting disabilities or even fatal outcomes. This narrative review aims to provide a comprehensive overview of n-irAEs, focusing on their recognition and management. The review addresses a spectrum of n-irAEs, encompassing myositis, myasthenia gravis, various neuropathies, and central nervous system complications, such as encephalitis, meningitis, and demyelinating diseases. The key features of n-irAEs are emphasized in this review, including their early onset after initiation of ICIs, potential association with non-neurological irAEs and/or concurrent oncological response, the significance of ruling out other etiologies, and the expected improvement upon discontinuation of ICIs and/or immunosuppression. Furthermore, this review delves into considerations for ICI re-challenge and the intricate nature of n-irAEs within the context of pre-existing autoimmune and paraneoplastic syndromes. It underscores the importance of a multidisciplinary approach to diagnosis and treatment, highlighting the pivotal role of severity grading in guiding treatment decisions.

## 1. Introduction

The immune system detects and eradicates cancer cells by recognizing tumor-associated antigens (neoantigens) captured by antigen-presenting cells (APCs) and presented to T cells via the major histocompatibility complex (MHC) I. Positive regulatory signals for T-cell activation include the binding of CD80 on APCs to CD28 on T cells, activating various signaling cascades, such as the phosphoinositol-3-kinase (PI3K)—protein B (AKT)—mammalian target of rapamycin (mTOR) pathway, the mitogen-activated pathway kinase (MAPK), and the Janus kinase pathway, ultimately leading to NF-kβ activation and IL-2 production. However, cancers often evade immune detection through immune checkpoints, such as the cytotoxic T lymphocyte-associated protein 4 (CTLA-4), the programmed cell death protein-1 (PD-1), and the programmed cell death ligand-1 (PD-L1), which inhibit the physiological activation of T cells. CTLA-4 competes with CD28 and binds to B7 on APCs, impeding T-cell activation in the priming phase. Similarly, PD-1 on T cells suppresses their activation in the effector phase by downregulating T-cell receptor surface expression and hindering MHC recognition. Moreover, PD-1 inhibits T-cell proliferation by blocking the PI3K-AKT-mTOR cascade and suppressing cyclin-dependent kinases (CDKs) [1,2,3]. Figure 1 describes the action mechanisms of ICIs.

Immune checkpoint inhibitors (ICIs) disrupt inhibitory signals, reactivating the immune system. Notably, ipilimumab and tremelimumab target CTLA-4, while atezolizumab, durvalumab, and avelumab target PD-L1, and nivolumab and pembrolizumab target PD-1 [4]. In current clinical practice, ICIs are used across various cancer types employing different strategies and durations. Initially evaluated in the metastatic setting, they can be used for an indefinite period as monotherapy, in combination with chemotherapy, or with tyrosine kinase inhibitors. Furthermore, they have shown efficacy in the perioperative setting with a limited duration period. A combination therapy involving anti-CTLA-4 and anti-PD-1 inhibitors is also proposed in metastatic cancer, with dosage and duration varying depending on the cancer type [4,5]. A large meta-analysis demonstrated that most patients treated with ICIs experience immune-related adverse events (irAEs); this irAEs rate can reach 83% with anti-CTLA-4, 72% with anti-PD-1, and 59% with anti-PD-L1 therapy [6]. Neurotoxicities associated with ICIs, referred to as neurological immune-related adverse events (n-irAEs), are relatively rare compared to other irAEs, such as skin or gastrointestinal toxicities. A review of 59 studies, encompassing a total of 9,208 patients, revealed an incidence of n-irAEs reaching 3.8% with anti-CTLA4 therapy, 6.1% with anti-PD1 therapy, and 12% with combination therapy (i.e., anti-CTLA4 plus anti-PD1) [7]. However, this incidence is probably over- or under-estimated, as 55% of these n-irAEs consisted in non-specific symptoms, such as asthenia or headache and confusion, which are not included in a recent classification of n-irAEs [8,9,10]. Grade ≥ 3 n-irAEs represented less than 2% of these cases [11]. 

The early recognition of n-irAEs is crucial, as their potential severity is significant. With a mortality rate that can reach 24%, these toxicities generate the most frequent fatalities related to ICIs, alongside myocarditis [12,13]. This is particularly true for conditions such as encephalitis, Guillain–Barré syndrome (GBS), and myasthenia gravis (MG), which lead to rapid deterioration and have notably high mortality rates [10,12,13]. A multidisciplinary approach is thus essential to ensure a rapid diagnosis and optimal management.

The aim of this narrative review is to provide a comprehensive overview and update on these conditions, which often represent a challenge for clinicians. This review draws mainly upon the latest guidelines from the European Society for Medical Oncology (ESMO), the American Society of Clinical Oncology (ASCO), and the Society for Immunotherapy of Cancer (SITC) [14,15,16].

## 2. General Management

### 2.1. Classification

Neurological disorders are divided in two categories: central and peripheral disorders. Central nervous system (CNS) disorders, representing 17–25% of n-irAEs, include encephalitis/encephalopathy, meningitis and demyelinating conditions such as multiple sclerosis and transverse myelitis [8,10,13]. 

Peripheral nervous system disorders account for 75–83% of cases, include neuropathies, polyradiculoneuropathies (such as GBS and its variants), neuromuscular junction dysfunctions (such as MG) and myopathy/myositis [8,10,13]. 

### 2.2. Pathogenesis

Numerous mechanisms may be advanced for irAE development; yet, the pathogenesis remains largely unknown. The intricacies of the immune system, coupled with the low incidence of n-irAEs and limited access to histological samples (particularly CNS tissue), hinder our understanding of these autoimmune toxicities. 

The mechanism of action of ICIs is not specific to tumoral tissue, targeting molecules such as CTLA-4 and PD-1 expressed on various immune cells beyond T cells, including regulatory T cells (Tregs) and B cells. Tregs play protective roles, preventing excessive immune activation and impairing autoimmune neurological disorders, while B cells produce antibodies after immune stimulation. ICI therapy disrupts these immune properties, leading to a disequilibrium in immune self-modulation. This may result in decreased peripheral self-tolerance due to Treg depletion and excessive autoantibody production through B-cell activation.

Another potential mechanism underlying irAEs is molecular mimicry, where the similarities between host antigens and tumor antigens induce a cross-reactive autoimmune response. For example, in melanoma patients developing ICI-related demyelinating polyneuropathy, molecular mimicry between melanocytes and Schwann cells may occur due to shared epitopes for immune responses. This phenomenon is akin to paraneoplastic syndromes (PNS), where ICIs may promote immune-mediated PNS via cross-reactivity against self-antigens expressed on both neural cells and tumors [17].

Additionally, ICIs can affect non-hematopoietic cell lines expressing the target immune checkpoint ligands. For instance, the high rate of immune-related hypophysitis observed during anti-CTLA4 therapy may be attributed to CTLA-4 protein expression on pituitary endocrine cells [18]. Moreover, PD-1 protein expression has been observed in the cortex and basal ganglia, with RNA expression of PD-1, PD-2, and CTLA-4 verified throughout the CNS [17].

ICIs may exacerbate pre-existing immune responses against the nervous system, such as multiple sclerosis, auto-immune encephalitis, inflammatory demyelinating neuropathies, and MG. This escalation of local inflammation or unveiling of latent central or peripheral inflammation is challenging to confirm, as clinical trials involving ICIs typically exclude patients with autoimmune diseases due to their potential for serious irAEs, making robust clinical responses difficult to establish.

The reasons why some patients develop irAEs while others do not, remain unclear. Genetic factors, such as CTLA-4 polymorphisms, have been associated with increased auto-immune disease risk, but studies have not consistently linked specific genotypes to irAE susceptibility [19]. Similarly, the role of microbiome composition in irAE development is uncertain, although some evidences suggest correlations between certain gut bacteria and resistance to immune-related colitis during ipilimumab treatment for melanoma. However, further research—particularly prospective studies—is necessary to elucidate the relationships between genetic predisposition, microbiome composition, and irAE susceptibility, especially in the context of n-irAEs [20,21].

While n-irAEs share similar features with their traditional counterparts, it is important to be aware of significant differences, which contribute to accurate diagnosis and enable tailored treatments. For instance, ICI-induced neuropathies may behave similarly to classical GBS but are more likely to respond to corticosteroids and often present with lymphocytic pleocytosis in cerebro-spinal fluid (CSF) [7,10]. Figure 1 describes some hypotheses for the development of (neurological) irAEs. 

### 2.3. Approaching the Diagnosis

A multidisciplinary approach is of importance, involving neurologists but also other specialists (ophthalmologists, rheumatologists, radiologists, cardiologists…) in order to provide the most appropriate diagnosis and treatment. 

The diagnosis of n-irAEs first highlights the importance of systematically excluding other potential causes before attributing symptoms to n-irAEs. These include a wide range of conditions, such as infections (viral or bacterial), cancer progression (including leptomeningeal carcinomatosis), metabolic issues (hypothyroidism, hypercalcemia), nutritional deficiencies, neurotoxicity from other treatments (e.g., platinum-induced sensory neuronopathy and cognitive side effects from radiotherapy), epilepsy, and vascular causes. Other features can help to suspect irAEs: (a)Timing is crucial for identifying n-irAEs, as they usually emerge within 3–6 months after initiating ICI treatment, with a median onset of 4 weeks (ranging from 1 to 85 weeks) [9,22,23,24,25]. Neurological symptoms emerging beyond 6 months after the last ICI dose are less likely to be secondary to an irAE, although it cannot be entirely ruled out solely on this basis [8];(b)Clinical and/or radiological evidence of cancer control coinciding with the onset of neurological symptoms supports n-irAE diagnosis rather than attributing the symptoms to cancer progression [22];(c)The concurrent occurrence of neurological symptoms with other irAEs increases the likelihood of a n-irAE [11,22];(d)A positive response to ICI interruption and/or immunosuppression also suggests a n-irAE, although it is not a definitive indicator [8].

### 2.4. Diagnosis Tools

The optimal diagnostic evaluation for patients with suspected n-irAEs should be tailored based on their clinical presentation and may encompass different tools, including brain and/or spinal magnetic resonance imaging (MRI), lumbar puncture for CSF analysis, electroneuromyography (EMG), electroencephalography (EEG). Brain or meningeal biopsy, although rarely necessary, has to sometimes be considered to exclude alternative diagnosis (e.g., chronic pachymeningitis or persistent suspicion of leptomeningeal carcinomatosis despite negative lumbar puncture). Auto-immune antibodies belong to the common tools in the diagnosis of n-irAEs. Most are assessed in serum, whereas antibodies specifically targeting membrane components often require a CSF analysis (e.g., NMDAR, LGI1, CASPR2). Some, referred to as syndrome-associated autoantibodies, support the diagnosis of specific syndromes [8,26]. However, as many antibodies lack specificity, it is essential to correlate the clinical presentation with established antibody-associated syndromes before making a diagnosis [8]. Furthermore, given the prevalence of false positives, the results should be validated using confirmatory tests [22,27]. Antinuclear antibody (ANA) testing is non-specific but sometimes advised, as it can suggest an autoimmune tendency [8,14,15,16].

The diagnosis of a n-irAE also requires a broader evaluation; myositis/myopathy or MG should lead to investigations of myocarditis with EKG, troponin levels, and consideration of additional tests, such as brain natriuretic peptide (BNP), CK-MB, cardiac ultrasonography (US), and cardiac MRI [8]. Moreover, pulmonary function tests and video-fluoroscopic swallow studies can help in assessing restrictive syndromes and dysphagia secondary to MG, myositis, or GBS.

## 3. Importance of Rapid Diagnosis and Severity Score

A multidisciplinary approach is required, involving neurologists and other specialists, including ophthalmologists, cardiologist, pneumologist and rheumatologists in suspected ocular or muscular involvement cases [28]. This approach, linked to improved outcomes, helps locate the neurological lesion, rule out other causes, and guide both diagnostic evaluation and therapeutic management [29]. Therefore, consulting organ specialists is recommended for all severity levels [14,16]. Guidon et al. proposed a set of criteria with three levels of diagnostic certainty—definite, probable, and possible—based on clinical signs, radiological, electrophysiological, and biological findings, as well as on the evolution under immunomodulatory agents [8].

Severity grading is crucial for determining the appropriate therapeutic approach and level of care. The severity of n-irAEs has been categorized based on the Common Terminology Criteria for Adverse Events v5.0 (CTCAE) system, originally designed for chemotherapy-related toxicities, and it may not fully capture the severity of irAEs [14]. Guidon et al. introduced a novel severity-based grading system, essential for guiding treatment decisions. Table 1 summarizes the severity score of n-irAEs based on CTCAE v5.0 and Guidon et al.’s recommendations. 

It is important to note that for conditions such as MG, GBS, and encephalitis, Grade 1 may not be applicable due to the potential for rapid deterioration and the importance of early management. These patients should be admitted into units capable of rapid transfer to intensive care level monitoring, with regular assessment of bulbar, autonomic, or respiratory symptoms. A rapid and accurate diagnosis can only be achieved through collaboration with neurological teams experienced in managing n-irAEs. Physicians should also be vigilant about the potential impacts of ICIs on other organs, such as the cardiac system. Table 2 summarizes the severity score and related treatment for key n-irAE entities, including irMyopathy/Myositis (irM/M), irNeuromuscular Junction Disorders (irNMJD), irNeuropathy, irMeningitis, irEncephalitis, and irDemyelinating Diseases (irDDs).

## 4. Recognizing the Key Neurological Immune-Related Adverse Events

### 4.1. irMyopathy/Myositis (irM/M)

#### 4.1.1. General Overview

Immune-related M/M can either emerge de novo or as a reactivation of pre-existing paraneoplastic polymyositis or dermatomyositis [15]. It is the most frequent n-irAE, accounting for approximately 32% of cases [10,30]. Typically, the onset occurs within 2 months following ICI therapy [24,31], with a median of 5–6 weeks [24,30,31,32]. Anti-PD(L)-1 therapy is more likely associated with irM/M than anti-CTLA4 therapy [33,34]. 

Up to 70–80% of patients improve after discontinuing ICI therapy and starting immunomodulatory treatment [10]. However, half of patients experience sustained effects after 5 years, and the mortality rate can reach 17%, resulting mainly from bulbar involvement, respiratory muscle failure, and concurrent myocarditis [10,30]. 

The “triple M syndrome” describes the frequent overlap of myositis, myocarditis, and MG in ICI-treated patients, and it has been reported in up to 53% of myositis cases [35]. This syndrome is associated with poor prognosis, with the mortality rate reaching 60% [36]. Therefore, routine screening for myocarditis and MG is mandatory in all patients presenting with irM/M.

#### 4.1.2. Clinical Presentation

irM/M presents with a fixed—sometimes focal—muscular weakness [8]:-Proximal limb muscles are more affected than distal ones, leading to limitations in ambulating, raising, and lifting arms;-Axial weakness, particularly in the cervical region, results in difficulties in neck extension and flexion;-Oculo-bulbar involvement, characterized by ptosis, diplopia, dysphagia, and dysarthria, is a distinct and prominent feature of irM/M, being the primary or sole manifestation in 42% of patients, contrasting with those unexposed to ICI [37].

Muscular pain generally occurs only in severe myositis cases and sometimes precedes muscular weakness. Cutaneous manifestations associated with dermatomyositis (malar rash, heliotrope rash, Gottron papules, and shawl sign) are rare in de novo irM/M. 

#### 4.1.3. Diagnostic Tools

The following tests should be conducted:-Creatine kinase (CK) levels, while often elevated, do not consistently correlate with the severity of myopathy and can be normal in up to 30% of cases [37]. Aldolase, another marker of muscle breakdown, may be high even in the absence of CK elevation, providing additional diagnostic insight [37];-EMG to detect a myopathic pattern (although it presents as normal in up to 20% of cases) and to guide muscle biopsy [10,35,37];-EKG and troponin levels;-Assessment of MG-specific antibodies (AChR, MuSK, LRP4) can also help to evaluate the presence of co-occurring myasthenia, especially when ocular or bulbar symptoms are present [35,37]. While myositis-specific antibodies (e.g., ANA, Jo-1, PL-7, PL-12, EJ) may be considered for pretherapeutic muscular manifestations or suspected underlying dermatomyositis, they often yield negative results in ir-M/M [31,37,38];-Limb MRI with contrast typically reveals STIR hyperintensity or contrast enhancement in affected muscles. While these findings are non-specific and can be observed in denervated muscles, they provide diagnostic support and assist in directing biopsy [8].-Muscle biopsy is only recommended in cases of doubtful diagnosis [31,37];-Pulmonary function tests and video-fluoroscopic swallow studies to assess restrictive syndromes and dysphagia, respectively.

### 4.2. irNeuromuscular Junction Disorders (irNMJD), Including Mysthenia Gravis (MG)

#### 4.2.1. General Overview

Immune-related MG can present either as new-onset MG or as exacerbations of a pre-existing condition [39,40]. MG tends to develop particularly early, with the median onset of 4–6 weeks after starting ICI, and in some cases, as early as 6 days [24,32]. It is more likely to develop in patients receiving anti-PD-(L)1 therapy compared to those treated with anti-CTLA-4 therapy [24]. While most patients improve, the mortality rate can reach up to 28–30%, primarily due to complications, such as bulbar involvement, respiratory muscle failure, and concurrent myocarditis [10,41]. Long-term immunosuppression is often necessary, except in a few patients who experience minor manifestations confined to ocular or facial muscles [14]. Distinguishing between MG and myositis can be challenging, as myositis can mimic MG, leading to a clinical presentation referred to as “pseudo-myasthenia” [8,42,43]. In such cases, myopathy can display weakness patterns in the oculo-bulbar, respiratory, and axial muscles. Furthermore, AChR antibodies can be non-specifically present in patients with myopathy, but they often do not show clinical or electrophysiological fluctuations characteristic of neuromuscular junction dysfunction [37]. Instead, these patients frequently exhibit varying degrees of myopathy, ranging from elevated CK levels to necrotizing myopathy. About 80% of patients also experience non-neurological irAEs, such as myocarditis [32]. 

#### 4.2.2. Clinical Presentation

Neuromuscular junction disorder should be suspected if a patient exhibits fluctuating, exercise-dependent muscle weakness, which usually involves

-Proximal and axial cervical muscles (neck and shoulder weakness) [16];-Ocular muscles (diplopia, asymmetrical ptosis, and/or fatigability);-Bulbar muscles (dysphagia, dysphonia, and dysarthria);-Respiratory muscles, affecting over 50% of cases, more frequently than in idiopathic MG [10,32].

#### 4.2.3. Diagnostic Tools

The ice pack test is a simple yet helpful diagnostic tool [44]: placing an ice pack on the ptotic eyelid for two minutes should relieve symptoms in the case of MG, likely due to the slowing of cholinesterase activity.

The recommended assessments include

-Electrodiagnostic studies to detect a decremental pattern (reduction in the amplitude of action potentials with repeated nerve stimulation), a hallmark of MG [45];-AChR antibodies, detected in 59–66.7% of patients [10,41];-Consideration of MuSK and LRP4 antibodies, especially in those with pre-existing NMJ disorders, due to their rare documentation [41];-Myocarditis and myositis evaluation via EKG, troponin, CK, or aldolase, given their frequent association with irNMJD;-MRI of the brain, spinal cord, or orbit to exclude cancer involvement of CNS.-Pulmonary function tests and fluoroscopic swallow evaluation for patients with respiratory symptoms or dysphagia;-TSH levels, given the potential impact of thyroid disorders on neuromuscular function.

### 4.3. irNeuropathy, Including Guillain–Barré Syndrome

#### 4.3.1. General Overview

Neuropathies following ICI therapy encompass a broad spectrum of phenotypes, including various forms of polyradiculoneuropathies, such as GBS and rarer variants (acute inflammatory demyelinating polyneuropathy (AIDP), Miller Fisher syndrome (MFS), acute motor axonal neuropathy (AMAN), acute motor and sensory axonal neuropathy (AMSAN), chronic inflammatory demyelinating polyneuropathy (CIDP)) [8,10,46]. The incidence of irGBS is less than 0.5%, typically occuring within 3–4 months of therapy initiation, although delayed onset is possible [14,24,47]. irGBS is likely related to anti-CTLA-4 therapy and melanoma [10,24]. The mortality rate was reported to reach 11% [10]. Compared with idiopathic GBS, irGBS rarely occurs after an infectious episode, may present with CSF pleocytosis, and may have a better response to corticosteroids [10,14,15]. ICI-induced neuropathies may behave similarly to classical GBS but are more likely to respond to corticosteroids and often present with lymphocytic pleocytosis in CSF.

#### 4.3.2. Clinical Presentation

The diagnosis of irGBS is primarily clinical. The key features include rapidly progressive, symmetrical, ascending weakness associated with hyporeflexia and muscle atrophy. The weakness may range from mild walking difficulties to near-complete paralysis of limb, facial, respiratory, and bulbar muscles. Sensory disturbances, such as paresthesias and neuropathic pain (usually in the lumbar and/or crural regions), along with autonomic dysfunction (blood pressure and heart rate fluctuations, urinary retention, ileus), are possible. Atypical symptoms, such as cranial nerve involvement, bulbar symptoms, and dyspnea may also be observed.

#### 4.3.3. Diagnostic Tools

The recommended studies include 

-Electrodiagnostic studies, which should demonstrate a demyelinating pattern indicative of acquired polyradiculoneuropathy, occasionally with secondary axonal loss [10,47,48]. Concomitant myopathy is rare;-CSF analysis, usually showing elevated protein levels and pleocytosis [10,47,48];-Anti-ganglioside and neuronal antibodies; while typically negative in ICI-induced cases, can help identify GBS variants (anti-GQ1b for MFS or BBE) and paraneoplastic neuropathies (anti-Hu, anti-amphiphysin), respectively [10,47];-A large infectious screening, as many infections can cause idiopathic GBS: CMV, mycoplasma, HIV, campylobacter, syphilis, Lyme…;-Brain and spine MRI, especially when bulbar weakness or quadriparesis are present, to rule out a compressive lesion and other causes. Root nerve thickening and enhancement can be present in ICI-related GBS [10,23,48].

### 4.4. irMeningitis

#### 4.4.1. General Overview

ICI-related meningitis has an incidence of less than 0.5% and usually occurs within 3–4 months of ICI initiation, after a median of 2 cycles (ranging from 1 to 14) [24,49,50]. Anti-CTLA-4 therapies are more likely to induce irMeningitis than anti-PD(L)-1 therapies [10,24,51]. Most cases fully recover without long-term consequences [49].

#### 4.4.2. Clinical Presentation

The symptoms may be aspecific, ranging from headache only to meningeal syndrome (headache, neck stiffness, photophobia, vomiting, with or without fever) [10]. The signs and symptoms of encephalitis (confusion, behavioral changes, aphasia, seizure-like activity, short-term memory disturbances, or other focal neurological deficits) are associated with a worse prognosis [14,43].

#### 4.4.3. Diagnostic Tools

The recommended diagnostic modalities include

-Blood and CSF samples to exclude bacterial and viral meningitis. CSF analysis usually reveals a lymphocytic pleocytosis and hyperproteinorrachia [50];-Brain MRI, which shows meningeal enhancement in half of patients [50].

### 4.5. irEncephalitis

#### 4.5.1. General Overview

irEncephalitis represents the most frequent CNS condition among n-irAEs, with an incidence of 0.5–1% [24,49]. It typically occurs within 3–4 months of ICI initiation, with a median onset of 61 days (ranging from 18 to 153 days), and it is more likely to manifest with anti-PD(L)-1 therapy and lung cancer [24]. Despite favorable outcomes in most cases, rapid deterioration may occur, leading to epilepsy, altered mental status, respiratory failure, and occasionally death [52,53].

#### 4.5.2. Clinical Presentation

irEncephalitis often presents with aspecific symptoms [43]. Two main presentation types have been described [42,43,52]:-Diffuse, meningoencephalitis-like picture, with general symptoms (fever, headache, altered consciousness, and/or seizures) and without focal signs;-Focal presentations, including limbic encephalitis (neuropsychiatric disturbances, memory issues, behavioral changes, delusions, hallucinations, temporal epilepsy), cerebellitis (ataxia, dysmetria), brainstem encephalitis (altered vigilance, cranial nerve paresis, vertigo, bulbar syndromes), basal ganglia encephalitis (extrapyramidal syndrome with abnormal movements), dysphasia, pyramidal syndrome, sensory disturbances, and dysautonomia.

#### 4.5.3. Diagnostic Tools

The following workup should be conducted:-Brain MRI with contrast. The most frequent findings are hyperintensities in the mesial temporal regions on T2/FLAIR sequences for limbic encephalitis and meningeal enhancement/thickening for meningoencephalitis [51]. MRI can be repeated, as radiological abnormalities are sometimes delayed [10,42,52,53,54,55];-Lumbar puncture often reveals lymphocytic pleocytosis and hyperproteinorrachia (77–98%), oligoclonal bands (38–53%), and normal or slightly raised glycorrhachia [52,53,54];-Neuronal antibody testing in blood and CSF. The seropositivity rate ranges from 6% to 54% [52,54]. Specifically, onco-neuronal antibodies targeting intracellular antigens are more common [52]. There have also been reports of other antibodies, including anti-GFAP in the context of meningoencephalitis, as well as anti-Ri, anti-GAD, anti-NMDAR, anti-CASPR2, anti-CRMP5, and anti-SOX1 [42,43]. The presence of these antibodies is often linked to an unfavorable prognosis;-EEG, which can indicate focal or diffuse slowing, subclinical seizures, or status epilepticus;-Additional assessments may involve spinal MRI, further infectious workup, and—in rare cases—brain biopsy to rule out oncological progression.

### 4.6. irDemyelinating Diseases (irDDs)

#### 4.6.1. General Overview

Demyelinating diseases represent a heterogeneous group of conditions affecting the myelin sheath of nerve fibers in the brain, spinal cord, and optic nerve. These include multiple sclerosis (MS), acute disseminated encephalomyelitis (ADEM), neuromyelitis optica spectrum disorder (NMOSD), optic neuritis (ON), and transverse myelitis (TM), among others [8,56]. The incidence of irDDs is less than 0.5% [24,49]. Manifestations typically appear at a median time of 6.5 weeks after ICI initiation (ranging from 1 to 43 weeks) [56]. Most cases resolve at least partially after ICI interruption [24,49].

#### 4.6.2. Clinical Presentation

The symptoms vary depending on the demyelinating lesions [8]:-Involvement of cerebral hemispheres can lead to muscle weakness with pyramidal signs, sensory disturbances, and mental status changes;-Posterior fossa localization is associated with diplopia, ophthalmoplegia, nystagmus, ataxia, dysmetria, dysarthria, and dysphagia;-Optic neuropathy results in reduced visual acuity, visual field loss, dyschromatopsia, afferent pupillary defect, and optic disc edema;-Transverse myelitis manifests as sensory disturbances with a sensory level, pyramidal weakness, and ataxia.

Other symptoms include numbness, paresthesia, and autonomic dysfunction (urinary and fecal incontinence). 

#### 4.6.3. Diagnostic Tools

The recommended workup includes

-Brain, orbit, and/or spinal MRI with contrast, which typically shows enhancements and/or hyperintense T2/FLAIR lesions, although no definitive imaging feature has been identified [43];-Lumbar puncture, usually revealing lymphocytic pleocytosis and elevated protein levels [56];-Autoantibody testing, including demyelinating antibodies (i.e., AQP4 and MOG) in serum and spinal fluid, although the impact on sensitivity is limited compared to serum analysis alone. While most patients test negative for these antibodies, AQP4, CRMP5, Hu, or other neural antibodies can be present in some patients [10,43,56,57];-Other diagnostic tools include EEG, neuro-ophthalmologic evaluation and evoked potentials;-Brain biopsy—exceptionally—to allow definitive proof of CNS demyelination.

## 5. Treatment

### 5.1. General Management

ICIs have become indispensable in cancer treatment, significantly enhancing the response rates, progression-free survival, and overall survival. In some cancers, ICIs represent the cornerstone of therapy, and in certain cases, the only viable option. It is crucial to acknowledge the absence of prospective, randomized studies guiding therapy in this context. The current recommendations are based on retrospective data and expert opinions alone, emphasizing the need to view the proposed regimens as suggestive rather than definitive [14]. 

According to certain guidelines, Grade 1 n-irAEs (characterized by mild symptoms, which do not disrupt daily activities) may permit the continuation of ICI treatment in select cases. However, it is essential to exercise caution, as a Grade 1 event has the potential to escalate to a Grade 2 severity level rapidly. Delaying administration may also be considered in order to confirm the diagnosis and ensure there is no deterioration. Of course, this has to be discussed with the patient, based on the oncological situation. 

To minimize the risk of exacerbating an infectious cause while addressing the potential delay in ruling out infectious meningitis, two strategies can be considered:(a)Prioritize excluding bacterial infections and—if feasible—viral infections before initiating immunosuppression;(b)Administer antimicrobials concurrently while awaiting negative PCR and culture results, particularly in severe cases.

### 5.2. Steroids, IVIG, and Plasmapheresis 

Table 2 summarizes the recommended management of key n-irAEs. 

As with most immune-induced toxicities, the initial treatment step often involves suspending the ICI and starting corticosteroid therapy [14,15]. A suitable dosage regimen is prednisolone (0.5–1 mg/kg) for Grade 2 symptoms and high-dose oral or intravenous prednisolone (1–2 mg/kg) for significant neurological toxicity (Grades 3 and 4). If no improvement occurs after the initial dosing, steroid doses can be escalated up to 2 mg/kg/day. However, for conditions such as MG, GBS, or encephalitis, it is prudent to consider higher doses, such as IV methylprednisolone at 1–4 mg/kg/day. Upon improvement, transitioning to oral steroids with a prolonged tapering over 4–8 weeks is suggested, with consideration of steroid-related adverse events, such as gastritis and osteoporosis. Re-assessment is warranted after 3–5 days, and pulse dose methylprednisolone (500–1000 mg/day for 3–5 days) has to be considered in case of deterioration.

Intravenous immunoglobulins (IVIG; 2 mg/kg/day over 3–5 days) or plasmapheresis (or plasma exchange; 5–7 sessions) should be considered in case of corticosteroid resistance, higher severity at onset, concurrent myocarditis, all cases of GBS, and the majority of MG cases, particularly in the presence of bulbar and/or respiratory involvement. The choice between IVIG and plasmapheresis should be individualized: IVIG are more readily available and require less monitoring compared to plasmapheresis, which should be favored in more acute situations due to their faster onset of action, possibly linked to the rapid clearance of autoantibodies, cytokines, and the ICI itself [58]. Contra-indications for plasmapheresis include renal failure, hypercoagulable states, sepsis, and hemodynamic instability, whereas IVIG should not be administered to individuals with a high thromboembolic risk or severe hyponatremia. 

In refractory or recurrent cases, escalation of immunosuppression can include options such as abatacept, mycophenolate mofetil, tacrolimus, azathioprine, cyclophosphamide, Rituximab, anti-TNFα, and anti-IL6 therapies. Natalizumab—a monoclonal antibody, which inhibits leucocyte migration through the blood–brain barrier by blocking α4-integrin—has exhibited benefits in a case of steroid-refractory ICI-related limbic encephalitis [59].

### 5.3. ICI Re-Challenge

As with other irAEs, ICI re-challenge may lead to a recurrence of symptoms. Retrospective data show a 12–29% recurrence rate after re-exposure to ICI [60,61,62,63]; however, drawing conclusions is challenging, as clinical outcomes are not always reported.

Globally, ICI re-challenge in n-irAEs should be avoided; however, in cases where there are limited alternative therapies and where ICI remains crucial for oncological management, a re-challenge may need to be considered on a patient-specific basis. 

Based on the general considerations for irAEs, we can suggest that

-ICI re-challenge or continuation should always be discussed with the patient, and the risk/benefit balance should always be weighted. A close collaboration with neurologists specialized in n-irAEs is highly recommended;-Since irAEs may correlate with an oncologic response, assessing disease control before deciding to re-challenge or continue ICI therapy is crucial in evaluating the real benefit for the patient. Retrospective analysis of a cohort of 937 patients with melanoma treated with ICIs showed an association between the development of n-irAEs (n = 76, 8%) and longer survival (HR = 0.4, 95% CI 0.32–0.77) [63];-Reinitiating ICI can be considered if irAE severity did not exceed Grade 1 or 2, the patient’s symptoms have resolved or at least stabilized, and corticosteroids have been reduced to a dose of less than 10 mg/day of prednisone [16]. However, particularly in Grade 2, the oncological situation should be re-evaluated in order to confirm the need to reinitiate ICI.-However, for severe symptoms (Grades 3 and 4) or conditions such as MG, GBS, or transverse myelitis, permanent ICI discontinuation is advised [16].

### 5.4. Pre-Existing Autoimmune and Paraneoplastic Neurological Syndromes (PNSs) 

ICI may also function as a trigger of both autoimmune and paraneoplastic neurological syndromes. Retrospective studies suggest that ICI may exacerbate pre-existing autoimmune neurological conditions, such as myositis, MG, and multiple sclerosis (MS), with the observed degradation rates shown to reach 71% and 60%, respectively [64,65,66]. Consequently, a risk/benefit assessment for patients with pre-existing immune-related neurological disorders is warranted before starting ICI. The use of some specific questionnaire (for instance, the one devised by Aoun et al.) and close collaboration with a neurological team dedicated to ICI complications are essential [66]. 

The impact of ICI on PNS is less clear. Retrospective studies indicate an increased risk of both exacerbation and de novo emergence in patients undergoing ICI treatment [67,68,69]. In practice, given the frequent overlaps in diagnostic and therapeutic features, it is recommended to classify PNSs as Grade 3–4 n-irAEs and manage them accordingly [69]; however, PNSs associated with antibodies targeting intracellular neuronal antigens tend to have a poorer response to immunosuppressive therapies compared to n-irAEs. 

## 6. Conclusions

Recognition of n-irAEs is not easy, as they encompass a broad spectrum of intricate neurological syndromes, often with non-specific heterogeneous presentations and incomplete overlap relative to their traditional counterparts. This complexity is further amplified in the context of pre-existing autoimmune and paraneoplastic syndromes. Rapid recognition is the key, as n-irAEs carry a significant risk of lasting disability and potentially fatal outcomes compared to other irAEs. 

The key indicators of n-irAE diagnosis include a close temporal relation with ICI initiation, concurrent irAEs, a simultaneous oncological response, and a clinical improvement after ICI discontinuation and/or introduction of immunosuppressive therapy. 

Early recognition is of importance in this field, as n-irAEs can carry a significant risk of lasting disability and potentially fatal outcomes compared to other irAEs. The primary therapeutic strategy remains the suspension of ICI therapy and the use of corticosteroids, possibly supplemented with other immunomodulatory agents. More severe conditions, such as encephalitis, MG, and GBS, warrant more aggressive interventions.

Understanding the consequences of ICI re-challenge after an n-irAE is an area, which still requires further investigations, since prospective data are lacking. 

Naturally, these considerations are challenging due to the scarcity of prospective evidence and definite consensus among the guidelines available to clinicians. 

## Figures and Tables

**Figure 1 pharmaceuticals-17-00501-f001:**
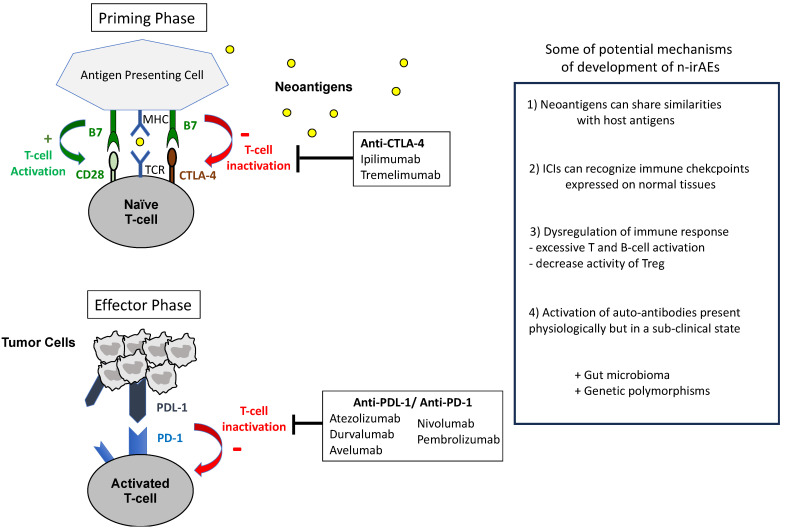
Mechanisms of action of ICIs and hypotheses for the development of (neurological) irAEs.

**Table 1 pharmaceuticals-17-00501-t001:** Global severity score of n-irAEs [8,14].

Severity Grade Based on CTCAE v5.0 (Guidon et al.’s Severity Equivalent)	Impact on Activities of Daily Life (ADL)
Grade 1 (Mild)	Symptoms do not interfere with ADLs or are subclinical
Grade 2 (Moderate)	Symptoms interfere with ADL and may require outpatient treatment
Grade 3 (Severe)	Symptoms interfere with ADLs and may require hospitalization for treatment
Grade 4 (Fulminant)	Life-threatening symptoms requiring emergent intervention (intubation, feeding tube)
Grade 5 (Death)	Death attributable to symptoms

**Table 2 pharmaceuticals-17-00501-t002:** Severity score for key n-irAEs and their respective management.

Condition	Severity	Management
irMyositis/Myopathy	Grade 1: Pauci-symptomatic increase in CK levels and/or asymptomatic weakness on neurological exam or mild myalgia without weakness.	Continue ICI therapy with close follow-up with regular swallowing and respiratory assessment.
	Grade 2: Mild-to-moderate weakness, including ocular weakness.	Discontinue temporary ICI and start oral prednisone (0.5–1 mg/kg/day) for at least 4 weeks, followed by a gradual tapering over 4–6 weeks. Permanently discontinue ICI in case of deterioration of symptoms despite steroids.
	Grade 3: Moderate-to-severe weakness in limbs or neck, altered walking or new need for an assistive device, bulbar symptoms (dysphagia, dysarthria, dysphonia), and/or dyspnea requiring monitoring.	Permanently discontinue ICI. Hospitalization is recommended, with IV methylprednisolone (1–2 mg/kg/day), followed by a gradual tapering regimen over 4–8 weeks, depending on symptom severity. Pulse methylprednisolone therapy (1 g/day for 5 days) may also be considered. In case of rapid progression with bulbar involvement or in case of persistent symptoms (after 7 days of steroids), start IVIG or plasmapheresis. After stabilization, continue with steroids (methylprednisolone at 1–2 mg/kg/day, followed by gradual tapering). In case of symptom persistence or recurrence despite steroids, consider methotrexate, azathioprine, mycophenolate mofetil, rituximab, anti-IL-6, or anti-TNF-α therapy. Formal contra-indication for ICI re-challenge.
	Grade 4: Respiratory weakness requiring intubation or non-invasive ventilation and/or dysphagia requiring feeding tube.	
irMyasthenia Gravis (MG)	Grade 1: Not applicable, given the potential for rapid deterioration and the importance of early management.	Discontinue temporary ICI and - start IV methylprednisolone (2–4 mg/kg/day), followed by a progressive tapering regimen over 4–8 weeks; - start oral pyridostigmine at a dosage of 30 mg three times a day, gradually increasing to a maximum of 120 mg four times a day, until symptom relief. IVIG or plasmapheresis can be considered. Globally, ICI re-challenge in MG should be avoided; however, a re-challenge may be considered in case of complete irAEs recovery after steroid arrest, progressive disease and limited therapeutic alternative.
	Grade 2: Mild-to-moderate weakness, including ocular weakness.	
	Grade 3: Moderate-to-severe weakness in limbs or neck, altered walking or new need for an assistive device, bulbar symptoms (dysphagia, dysarthria, dysphonia), and/or dyspnea requiring monitoring.	Definitively discontinue ICI. Start IV methylprednisone plus pyridostigmine (as below), and start IVIG or plasmapheresis. Pulse methylprednisone therapy (1 g/day for 5 days) may also be considered. For patients who do not respond, second-line therapies, such as rituximab, tacrolimus, or infliximab, should be considered. Formal contra-indication for ICI re-challenge.
	Grade 4: Respiratory weakness requiring intubation or non-invasive ventilation and/or dysphagia requiring feeding tube.	
irGuillain–Barré Syndrome (GBS)	Grade 1: Not applicable, given the potential for rapid deterioration and the importance of early management.	Discontinue ICI. Start IVIG or plasma exchange, and start IV methylprednisolone (2–4 mg/kg/day), with a gradual tapering regimen over 4–8 weeks. Globally, ICI re-challenge in irGBS should be avoided; however, a re-challenge may be considered in case of complete irAEs recovery after steroid arrest, progressive disease and limited therapeutic alternative.
	Grade 2: Any neuropathy-related weakness.	
	Grade 3: Walking impairment requiring assistive devices, bulbar symptoms (dysphagia, dysarthria, dysphonia), and respiratory dysfunction needing monitoring.	Definitively discontinue ICI. Start IVIG or plasma exchange, and start IV pulse methylprednisolone therapy (1 g/day for 5 days), with a gradual tapering regimen over 4–8 weeks. Formal contra-indication for ICI re-challenge.
	Grade 4: Respiratory weakness requiring intubation or non-invasive ventilation and/or dysphagia requiring feeding tube.	
irMeningitis	Grade 1: Headaches manageable with over-the-counter analgesics.	Discontinue ICI and resume ICI in case of Grade 1 confirmation and symptom recovery.
	Grade 2: Headaches requiring prescription analgesics.	Start oral prednisone (0.5–1 mg/kg/day) for a minimum of 2–4 weeks, followed by a gradual tapering regimen over 4–8 weeks. ICI re-challenge may be considered only in case of complete irAEs recovery after steroid arrest.
	Grade 3: Headaches necessitating intravenous analgesics or any corticosteroids; severe papilledema with associated visual deficit.	Definitively discontinue ICI. Start IV methylprednisolone at 1–2 mg/kg/day for a minimum of 2–4 weeks, with a gradual tapering regimen over 4–8 weeks. Methylprednisolone pulse therapy, IVIG, plasmapheresis, rituximab, and tacrolimus are potential treatments in refractory cases. Formal contra-indication for ICI re-challenge.
	Grade 4: Extra-ventricular drain for increased intracranial pressure.	
irEncephalitis	Grade 1: Not applicable, given the potential for rapid deterioration and the importance of early management.	Definitively discontinue ICI. Start IV methylprednisolone (1–2 mg/kg/day) with a gradual tapering regimen over 4–8 weeks. IVIG or plasmapheresis can be considered. Globally, ICI re-challenge in irEncephalitis should be avoided; however, a re-challenge may be considered in case of complete irAEs recovery after steroid arrest, progressive disease and limited therapeutic alternative.
	Grade 2: Subjective or mild cognitive deficits not significantly restricting daily activities.	
	Grade 3: Cognitive deficits limiting ADLs and/or seizures.	Definitively discontinue ICI. Consider ICU admission and start IV pulse methylprednisolone therapy (1 g/day for 5 days), followed by a gradual tapering regimen over 4–8 weeks and the addition of IVIG or plasmapheresis. In refractory situations, rituximab may also be used. Formal contra-indication for ICI re-challenge.
	Grade 4: Status epilepticus.	
irDDs (immune-related demyelinating diseases)	Grade 1: Asymptomatic or experiencing mild symptoms.	Discontinue temporary ICI in asymptomatic cases in order to exclude rapid deterioration.
	Grade 2: Mild unilateral visual changes, diplopia, and limitations in instrumental ADL due to deficits.	Discontinue ICI. Start oral or IV prednisolone (1–2 mg/kg/day), followed by a gradual tapering regimen over 4–8 weeks. Globally, ICI re-challenge in irDDs should be avoided; however, a re-challenge may be considered in case of complete irAEs recovery after steroid arrest, progressive disease and limited therapeutic alternative.
	Grade 3: Severe unilateral or bilateral vision loss, acute urinary retention, limitations in basic ADL.	Definitively discontinue ICI and start IV methylprednisolone pulse therapy (1 g/day for 5 days), followed by a gradual tapering regimen over 4–8 weeks. Consider addition of IVIG and plasmapheresis, or rituximab in case of symptom persistence or worsening.
	Grade 4: Intubation required due to respiratory failure caused by cervical or brainstem lesions.	

n-irAEs = neurological immune-related adverse events; ADL = activities of daily life; ICIs = immune checkpoint inhibitors; IVIG = intravenous immunoglobulins; MG = myasthenia gravis; GBS = Guillain–Barré syndrome; irDDs = immune-related demyelinating diseases.

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
