# Peer review of "Neurological Adverse Events Related to Immune Checkpoint Inhibitors: A Practical Review"

_pharmaceuticals, 2024, doi:10.3390/ph17040501_

Round 1
Reviewer 1 Report
Comments and Suggestions for Authors
François Zammit and Emmanuel Seront submitted a review about the n-irAEs of ICIs. The topic was interesting, and might arouse a certain impact in its field. Overall, the manuscript fell within the scope of Pharmaceuticals. However, a Major Revision must be conducted before a final decision for this paper. Please refer to the following comments:
1- The Abstract contained less than 130 words. Please consider to expand to ~200 words to showcase more valuable information of this review.
2- In the Introduction Section, please state the importance of n-irAEs detection.
3- Please try to name several ICIs, and show some information of them (e.g., molecular structure, dosage, targets, etc.).
4- References or sources for Table 1 must be cited.
5- An important issue: Seemingly, the current review only re-arrange the content from the guidelines from ESMO, ASCO and SITC, without further analysis and discussion. Please add a separate section to convey the authors’ personal opinion before the Conclusion Section, particularly about the implications for the clinic.
6- Going through the recent reviews published in Pharmaceuticals, a majority of them included high-quality figures. It was rather advisable to add some figures (either reused or self-prepared ones) in the revised version.
7- The format of References should be unified, especially the format of journals therein.
Author Response
- The Abstract contained less than 130 words. Please consider to expand to ~200 words to showcase more valuable information of this review.
 We adapted the abstract to reach 177 words
“Abstract: The increasing use of immune checkpoint inhibitors (ICI) in cancer therapy has brought attention to their associated neurotoxicities, termed neurological immune-related adverse events (n-irAEs). Despite their relatively rare incidence, n-irAEs pose a significant risk, potentially leading to severe, long-lasting disabilities, or even fatal outcomes. This narrative review aims to provide a comprehensive overview of n-irAEs, focusing on their recognition and management. The review addresses a spectrum of n-irAEs, encompassing myositis, myasthenia gravis, various neuropathies, and central nervous system complications such as encephalitis, meningitis, and demyelinating diseases. Key features of n-irAEs are emphasized in this review, including their early onset after initiation of ICIs, potential association with non-neurological irAEs and/or concurrent oncological response, the significance of ruling out other etiologies, and the expected improvement upon discontinuation of ICIs and/or immunosuppression. Furthermore, this review delves into considerations for ICI rechallenge and the intricate nature of n-irAEs within the context of pre-existing autoimmune and paraneoplastic syndromes. It underscores the importance of a multidisciplinary approach to diagnosis and treatment, highlighting the pivotal role of severity grading in guiding treatment decisions.”
- In the Introduction Section, please state the importance of n-irAEs detection.
We adapted the manuscript by incorporating this section
“The early recognition of n-irAEs is crucial as their potential severity is significant. With a mortality rate that can reach 24% [8], these toxicities generate the most frequent fatalities related to ICIs, alongside myocarditis [9]. This is particularly true for conditions like encephalitis, Guillain-Barré Syndrome (GBS), and myasthenia gravis (MG), which lead to rapid deterioration and have notably high mortality rates [6]. [6,8,9] A multi-disciplinary approach is thus essential to ensure a rapid diagnosis and optimal man-agement.”
- Please try to name several ICIs, and show some information of them (e.g., molecular structure, dosage, targets, etc.).
 Thank you for this comment; indeed, this part was lacking. We added this section describing the different ICIs, focusing on their mechanisms.
Line 32: “The immune system detects and eradicates cancer cells by recognizing tumor-associated antigens (neoantigens) captured by antigen-presenting cells (APCs) and presented to T cells via the major histocompatibility complex (MHC) I. Positive regulatory signals for T-cell activation include the binding of CD80 on APCs to CD28 on T cells, activating various signaling cascades such as the Phosphoinositol-3-kinase (PI3K) – Protein B (AKT) – mammalian target of Rapamycin (mTOR) pathway, the Mitogen Activated Pathway Kinase (MAPK) and the Janus Kinase Pathway, ultimately leading to NF-kβ activation and IL-2 production. However, cancers often evade immune detection through immune checkpoints like the cytotoxic T lymphocyte-associated protein 4 (CTLA-4), the programmed cell death protein-1 (PD-1), and the programmed cell death ligand-1 (PD-L1), which inhibit the physiological activation of T cells. CTLA-4 competes with CD28 and binds to B7 on APCs, impeding T-cell activation in the priming phase. Similarly, PD-1 on T cells suppresses activation by downregulating T-cell receptor surface expression and hindering MHC recognition. Moreover, PD-1 inhibits T-cell proliferation by blocking the PI3K-AKT-mTOR cascade and suppressing cyclin-dependent kinases (CDKs).
Immune checkpoint inhibitors (ICIs) disrupt inhibitory signals, reactivating the immune system. Notably, ipilimumab and tremelimumab target CTLA-4, while atezolizumab, durvalumab, and avelumab target PD-L1, and nivolumab and pembrolizumab target PD-1 [1]. In current clinical practice, ICIs are used across various cancer types employing different strategies and durations. Initially evaluated in the metastatic setting, they can be used for an indefinite period as monotherapy, in combination with chemo-therapy, or with tyrosine kinase inhibitors. Furthermore, they have shown efficacy in the perioperative setting with a limited duration period. Combination therapy involving anti-CTLA-4 and anti-PD-1 inhibitors is also proposed in metastatic cancer, with dosage and duration varying depending on the cancer type [1]. A large meta-analysis demonstrated that most patients treated with ICI experience immune-related Adverse Events (irAEs), reaching a rate of 83% with anti-CTLA-4, 72% with anti-PD-1, and 59% with an-ti-PD-L1 [2]. Neurotoxicities associated with ICI, referred to as neurological immune-related adverse events (n-irAEs), are relatively rare compared to other irAEs such as skin or gastrointestinal toxicities. A review of 59 studies, encompassing a total of 9,208 patients, revealed an incidence of n-irAEs reaching 3.8% with anti-CTLA4 therapy, 6.1% with anti-PD1 therapy, and 12% with combination therapy (i.e., anti-CTLA4 plus an-ti-PD1) [3]. However, this incidence is probably over or underestimated as 55% of these n-irAEs consisted of non-specific symptoms such as asthenia or headache and confusion, which are not included in a recent classification of n-irAEs [4–6]. Grade ≥3 n-irAEs rep-resent less than 2% of these cases [7].”
 We also added paragraph on pathogenesis
Line 91: Pathogenesis. “Numerous mechanisms may be advanced for irAE development, yet the pathogenesis remains largely unknown. The intricacies of the immune system, coupled with the low incidence of n-irAEs and limited access to histological samples (particularly CNS tissue), hinder our understanding of these autoimmune toxicities. The mechanism of action of ICIs is not tissue antigen-specific, targeting molecules such as CTLA-4 and PD-1 expressed on various immune cells beyond T-cells, including Regulatory T-cells (Tregs) and B-cells. Tregs play protective roles, preventing excessive immune activation and impairing autoimmune neurological disorders, while B-cells produce antibodies after immune stimulation. Tregs play protective roles in preventing excessive immune activation and autoimmune neurological disorders, while B-cells produce antibodies upon immune stimulation. ICI therapy disrupts these immune properties, leading to a disequilibrium in immune self-modulation. This may result in decreased peripheral self-tolerance due to Treg depletion and excessive autoantibody production through B-cell activation. Another potential mechanism underlying irAEs is molecular mimicry, where similarities between host antigens and tumor antigens induce a cross-reactive autoimmune response. For example, in melanoma patients developing ICI-related demyelinating polyneuropathy, molecular mimicry between melanocytes and Schwann cells may occur due to shared epitopes for immune responses. This phenomenon is akin to paraneoplastic syndromes (PNS), where ICIs may promote immune-mediated PNS via cross-reactivity against self-antigens expressed on both neural cells and tumors.
Additionally, ICIs can affect non-hematopoietic cell lines expressing the target immune checkpoint ligands. For instance, the high rate of hypophysitis observed with anti-CTLA4 may be attributed to CTLA-4 protein expression on pituitary endocrine cells. Moreover, PD-1 protein expression has been observed in the cortex and basal ganglia, with RNA expression of PD-1, PD-2, and CTLA-4 verified throughout the CNS.
ICIs may exacerbate pre-existing immune responses against the nervous system, such as multiple sclerosis, autoimmune encephalitis, inflammatory demyelinating neuropathies, and MG. This escalation of local inflammation or unveiling of latent central or peripheral inflammation is challenging to confirm, as clinical trials involving ICIs typically exclude patients with autoimmune diseases due to their potential for serious irAEs, making robust clinical responses difficult to establish.
The reasons why some patients develop irAEs while others do not remain unclear. Genetic factors, such as CTLA-4 polymorphisms, have been associated with increased autoimmune disease risk, but studies have not consistently linked specific genotypes to irAE susceptibility. Similarly, the role of microbiome composition in irAE development is uncertain, although some evidence suggests correlations between certain gut bacteria and resistance to immune-related colitis during ipilimumab treatment for melanoma. However, further research, particularly prospective studies, is necessary to elucidate the relationships between genetic predisposition, microbiome composition, and irAE susceptibility, especially in the context of n-irAEs, where evidence is lacking.
While n-irAEs share similar features with their traditional counterparts, it is important to be aware of significant differences which contribute to accurate diagnosis and enable tailored treatments. For instance, ICI-induced neuropathies may behave similarly to classical GBS but are more likely to respond to corticosteroids and often present with lymphocytic pleocytosis in cerebro-spinal fluid (CSF) [3,6].
- References or sources for Table 1 must be cited.
 Table 1 was deleted based on comment from Reviewer 2
- An important issue: Seemingly, the current review only re-arrange the content from the guidelines from ESMO, ASCO and SITC, without further analysis and discussion. Please add a separate section to convey the authors’ personal opinion before the Conclusion Section, particularly about the implications for the clinic.
 Thank you for this constructive comment. We completely rearranged the manuscript conception and improved our “general therapeutic approach” , with more clinical practice recommendations.
The specific therapeutic approach is inserted in the Table 3, in which the severity score with the related treatment is proposed.
- Going through the recent reviews published in Pharmaceuticals, a majority of them included high-quality figures. It was rather advisable to add some figures (either reused or self-prepared ones) in the revised version.
 We insert a figure summarizing the action mechanism of ICIs and the pathogenesis
- The format of References should be unified, especially the format of journals therein.
The reference list was modified.

Reviewer 2 Report
Comments and Suggestions for Authors
The authors conducted a detailed review of Neurological adverse events induced by immune checkpoint inhibitors. However, several points need revision and improvement.
My comments are listed below.
Major comments:
1. The Grades based on CTCAE criteria for the events the authors focus on should be indicated at the beginning of the paper (e.g., 2. General management). The authors present grades in the text from Section 3, which is very hard to read. Please consider revising this.
2. In Section "2.6 ICI rechallenge" of the paper, the authors discuss the re-administration of ICIs. Regarding central nervous system irAEs (such as encephalitis), re-administration of ICIs is not generally practiced. The authors' suggestion is very ambiguous and could lead to misunderstandings among readers. If the authors intend to mention the re-administration of ICIs in this review, it should be described in more detail.
3. The content of the tables and figures created by the authors needs to be more precise. For example, are Tables 1 and 2 essential information for the authors' paper? Additionally, while Table 3 shows three types of diagnoses, are "Probable diagnosis" and "Possible diagnosis" essential information? I believe it would be beneficial for clinicians to have tables showing treatment options and the possibility of re-administration for each grade of adverse neurological events. The authors' tables are generally ambiguous and contain unnecessary information. I recommend reconsidering all the figures and tables throughout the paper.
Author Response
- The Grades based on CTCAE criteria for the events the authors focus on should be indicated at the beginning of the paper (e.g., 2. General management). The authors present grades in the text from Section 3, which is very hard to read. Please consider revising this.
 We completely modified the manuscript with important adaptation based on the reviewers suggestions. This helps to clarify the text.
- In Section "2.6 ICI rechallenge" of the paper, the authors discuss the re-administration of ICIs. Regarding central nervous system irAEs (such as encephalitis), re-administration of ICIs is not generally practiced. The authors' suggestion is very ambiguous and could lead to misunderstandings among readers. If the authors intend to mention the re-administration of ICIs in this review, it should be described in more detail.
 We adapted our statement with this paragraph. We highlight the fact that ICI rechallenge should be avoided in case of n-irAE. However rechallenge may be considered in very selected patients.
As with other irAEs, ICI rechallenge may lead to a recurrence of symptoms. Ret-rospective data show a 12-29% recurrence rate after re-exposure to ICI [22,57–59] [22]; however, drawing conclusions is challenging as clinical outcomes are not always reported.
Globally, rechallenge of ICI in n-irAEs should be avoided; however, in cases where there are limited alternative therapies and where ICI remains crucial for oncological man-agement, rechallenge may need to be considered on a patient-specific basis..
Based on general considerations for irAEs, we can suggest that:
- Rechallenge or continuation of ICI should always be discussed with the patient and risk/benefit balance should always be weighted. A close collaboration with neurologists specialized in n-irAEs is highly recommended.
- Since irAEs may correlate with an oncologic response, assessing disease control before deciding to rechallenge or continue ICI therapy is crucial to evaluate the real benefit for the patient. The retrospective analysis of a cohort of 937 patients with melanoma treated with ICIs showed an association between the devel-opment of n-irAEs (n=76, 8%) and longer survival (HR = 0.4, 95% CI 0.32-0.77).[59]
- Reinitiating ICI can be considered if irAE severity did not exceed grade 1 or 2, the patient's symptoms have resolved or at least stabilized, and corticosteroids have been reduced to a dose of less than 10 mg/day of prednisone [12]. But, particularly in grade 2, the oncological situation should be re-evaluted in order to confirm the need to reinitiate ICI and this has to be hardly discussed with the patient. However, for severe symptoms (grades 3 and 4) or conditions like MG, GBS, or transverse myelitis, permanent ICI discontinuation is advised.
- The content of the tables and figures created by the authors needs to be more precise. For example, are Tables 1 and 2 essential information for the authors' paper? Additionally, while Table 3 shows three types of diagnoses, are "Probable diagnosis" and "Possible diagnosis" essential information? I believe it would be beneficial for clinicians to have tables showing treatment options and the possibility of re-administration for each grade of adverse neurological events. The authors' tables are generally ambiguous and contain unnecessary information. I recommend reconsidering all the figures and tables throughout the paper.
 We deleted the table 1 with differential diagnosis; we also deleted the table 2 with autoantibodies. In place, we added Table 1: CTCAE and Guidon score. We also added table 2 which summarizes the severity and related treatment for key n-irAEs. We think that this may underscore the treatment we have to offer for each severity score.

Reviewer 3 Report
Comments and Suggestions for Authors
Zammit and Seront write a comprehensive review article on the neurological irAEs associated with ICIs.
Overall, they give a very straightforward method of presenting the different irAEs in terms of general overview, clinical presentation, diagnostic tools, severity/grading, and treatment.
This would probably be okay for a journal of this caliber, although I would be interested in seeing more elaboration of retrospective studies with each subtype of n-irAEs.
Comments on the Quality of English LanguageMinor punctuational issues.
Author Response
Zammit and Seront write a comprehensive review article on the neurological irAEs associated with ICIs.
Overall, they give a very straightforward method of presenting the different irAEs in terms of general overview, clinical presentation, diagnostic tools, severity/grading, and treatment.
This would probably be okay for a journal of this caliber, although I would be interested in seeing more elaboration of retrospective studies with each subtype of n-irAEs.
 Thank you for this comment
We adapted our manuscript and try to be more clear and practical with easy description of the different strategies that can be proposed based on the severity score. We try alos to adjust our position concerning the rechallenge of ICI.

Round 2
Reviewer 1 Report
Comments and Suggestions for Authors
Thanks for your revision.
Reviewer 2 Report
Comments and Suggestions for Authors
The authors responded appropriately to my request for revisions.